# Boosting Semi-supervised Crowd Counting
# with Scale-based Active Learning

## ABSTRACT

The core of active semi-supervised crowd counting is the sample selection criteria. However, the scale factor has been neglected in active learning approaches despite the fact that the scale of heads varies drastically in the crowd images. In this paper, we propose a simple yet effective active labeling strategy to explicitly select informative unlabeled images, guided by the intra-scale uncertainty and inter-scale inconsistency metrics. The intra-scale uncertainty is quantified through the sum of the query-level entropy of images at different scales. Images are initially ranked based on this uncertainty for preselection. Inter-scale inconsistency is measured by the divergence between the query-level predictions of upscaled and downscaled images, allowing for the identification of the most informative images exhibiting the highest inconsistency. Additionally, we implement a progressive updating scheme for the semi-supervised crowd counting framework, in which the pseudo-labels for unlabeled images are refined iteratively. It further improves the counting accuracy. Through extensive experiments on widely used benchmarks, the proposed approach has demonstrated superior performance compared to previous state-of-the-art semi-supervised and active semi-supervised crowd counting methods.

## CCS CONCEPTS

• **Computing methodologies** → **Computer vision tasks**; *Active learning settings*; *Semi-supervised learning settings*.

## KEYWORDS

Crowd Counting, Active Learning, Semi-supervised Learning

## 1 INTRODUCTION

Recently, crowd counting has attracted attention due to its wide applications in real scenarios. Benefiting from the advancements in Deep Learning [13, 36], crowd counting approaches have made significant improvements in their performance. Nevertheless, most of the existing methods [27, 33, 40] are trained in a fully supervised manner, which poses challenges and burdens in the annotation process due to each individual in every image is required to be annotated.

To alleviate the annotation burden, numerous semi-supervised crowd counting methods [19, 24, 28, 38] have been proposed. These

*ACM MM, 2024, Melbourne, Australia*
© 2024 Copyright held by the owner/author(s). Publication rights licensed to ACM.
ACM ISBN 978-x-xxxx-xxxx-x/YY/MM
https://doi.org/10.1145/nnnnnnn.nnnnnnn

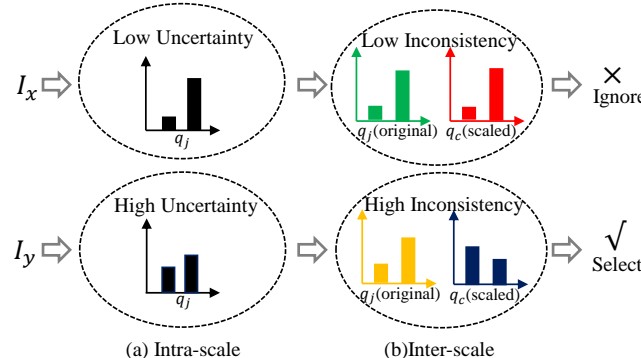

Figure 1: Scale-based Active Labeling strategy for crowd counting. The unlabeled image is selected for annotation with high uncertainty and inconsistency. (a) The entropy of query-level prediction measures the intra-scale uncertainty. (b) The divergence of query-level predictions indicates the inter-scale inconsistency.

methods typically involve the random selection of a subset of samples for annotation, followed by the use of both labeled and unlabeled images to train a crowd counting model. Some methods directly generate pseudo-labels for the unlabeled data [42], while others incorporate auxiliary tasks such as binary segmentation [24, 28], count ranking [23], and density prediction [19] to leverage the information contained in the unlabeled images.

Aware of the significant influence that the selection of image subset for annotation can have on performance, some researchers have proposed active semi-supervised crowd counting [25, 54] which actively annotates the most informative images or regions in the dataset. However, existing active semi-supervised methods are still far from fully-supervised methods. On the one hand, these methods only analyze images at the image-level and/or patch-level, but more fine-grained query-level analysis is not taken into account. On the other hand, scale is neglected despite the fact that head sizes significantly vary within the same image or between different images, which has been demonstrated as a key factor affecting the counting performance in the fully-supervised methods [3, 26].

To address the above issues, we propose a simple yet effective active labeling strategy to explicitly select informative unlabeled images for semi-supervised crowd counting guided by the intra-scale uncertainty and inter-scale inconsistency metrics, which is named Scale-based Active Learning(SAL). It offers fine-grained analysis from the query-level and takes scale as a main factor for data selection. The intra-scale uncertainty is quantified through the sum of the query-level entropy of images at different scales. The prediction probability of one query is shown on the left of Fig. 1. Images with higher uncertainty values are recognized as informative ones since the queries in them are hard to classify. Images are initially ranked

based on this uncertainty for preselection. Inter-scale inconsistency is measured by the divergence between the query-level predictions of upscaled and downscaled images. The query-level predictions are shown on the right of Fig. 1. A larger divergence indicates higher inconsistency of images that are sensible to scale. It allows the identification of the most informative images exhibiting the highest inconsistency. Additionally, we implement a progressive updating scheme for the semi-supervised crowd counting framework. In each iteration, a counting model is trained with the labeled and unlabeled by the supervision of ground-truth and pseudo-labels. Then the pseudo-labels are refined by this new model. With this scheme, the counting accuracy is further improved.

To verify the effectiveness of our proposed approach, we conducted comprehensive experiments on both the small datasets of ShanghaiTech_A [53], ShanghaiTech_B [53], UCF_QNRF [10], and large-scale datasets of JHU-Crowd++ [37] and NWPU-Crowd[46]. The results demonstrate that our method significantly surpasses previous semi-supervised and active semi-supervised crowd counting methods under the setting of 10% and 40% labeled data.

In a nutshell, the main contributions of the proposed active semi-supervised crowd counting model can be summarized as follows:

- We develop a simple yet effective strategy termed Scale-based Active Learning(SAL) for active labeling. The intra-scale uncertainty and inter-scale inconsistency are specifically designed to explicitly select informative unlabeled images.
- We propose a progressive updating scheme for the semi-supervised framework that progressively refines the pseudo labels.
- The proposed method achieves state-of-the-art performance on five datasets. It exceeds all the semi-supervised methods and active semi-supervised methods with 40% labeled data.

The rest of this paper is organized as follows: In Sec. 2, we review related research, including crowd counting with full annotation, crowd counting with limited annotation, and active learning. In Sec. 3, we introduce the preliminaries for active semi-supervised crowd counting, including settings and procedures. The details of our proposed approach are described in Sec. 4. Experimental results are presented and discussed in Sec. 5. Finally, we conclude and propose possible future work in Sec. 6.

## 2 RELATED WORK

### 2.1 Crowd Counting with Full Annotations

In recent years, fully-supervised crowd counting has gained satisfying performance based on the rapid development of deep learning. These methods can be divided into two categories: density-map based methods and point-matching methods.

The density map-based method was initially introduced in [14]. They utilized all point labels to generate a pseudo density map, which served as supervision for the predicted density map produced by the network. The estimated count is obtained by summing the values within the predicted density map. In recent years, numerous studies [9, 16, 53] have focused on density map schemes, addressing various challenges in crowd counting and pushing the boundaries of counting performance. Additionally, [12, 20, 30, 52] introduced attention networks to extract attentive features for boosting crowd counting. [3, 11] argued that scale variations largely influenced the

performance of crowd counting methods, and proposed adaptive scales or trellis architecture were employed to address it. In [27], the Bayesian assumption was introduced. Optimal transport was proposed in [44] to match the distributions. [4] incorporated probability maps into crowd counting and decoupled the task into two stages. In [29], a novel head size estimation method was proposed to reduce noise.

In addition to density map-based methods, several new solutions have been proposed. [40] proposed a purely point-based network that utilizes Hungarian Matching for point matching. Furthermore, [45] introduced two theoretically demonstrated criteria called Uniform Error Partition and Mean Count Proxies. Following the two criteria, the Uniform Error Partition Network is proposed. CLTR [18] proposed an end-to-end transformer network to directly predict the localization. To address instability, [32] proposed a local matching point-based framework. [22] viewed crowd counting as a decomposable point querying process and introduced PET to achieve dynamic processing of sparse and dense regions.

These fully supervised methods have demonstrated remarkable performance on diverse datasets. However, the process of annotating a crowd image in its entirety is both time-consuming and labor-intensive. Take UCF-QNRF dataset [10] as an example, the average annotation time for one image is over an hour. And annotators spent over 2000 hours on all of 1535 images.

### 2.2 Crowd Counting with Limited Annotations

Because the annotation process of crowd counting is time-consuming and complicated, an increasing number of methods have been proposed recently to achieve crowd counting with limited annotations.

In [23], a learning-to-rank framework that doesn't need manual annotation was proposed. Depending on the large collection of unlabeled crowd images, the framework was successfully trained to achieve crowd counting. Building upon this idea, a soft-label sorting network was proposed in [50] to directly regress the count, a challenging optimization task. In [48], binary ranking of image pairs was employed for network training. Additionally, [38] proposed a GP-based framework to exploit unlabeled data efficiently. Feature learning from unlabeled images was addressed in [24], where a generic feature extractor was trained. [28] propose a surrogate task to estimate the uncertain spatial regions and a differentiable transformation layer. In [19], a density agent is introduced to divide features into different groups and construct supervisions for unlabeled data. Furthermore, [49] proposed labeling only a patch of an image instead of the entire image. Furthermore, [1] presented a completely self-supervised crowd counting framework that only requires a few supervisions. Although [17] developed a counting method using count-level annotations, it does not reduce annotation effort. The optimal transport minimization (OT-M) algorithm was proposed in [21] and applied in semi-supervised crowd counting. Lastly, [15] introduced a supervised uncertainty estimation strategy to train the model using a surrogate function.

However, there is still a gap between the methods with limited annotation and full supervision. Moreover, the random selection of labeled images can significantly impact the model's performance.

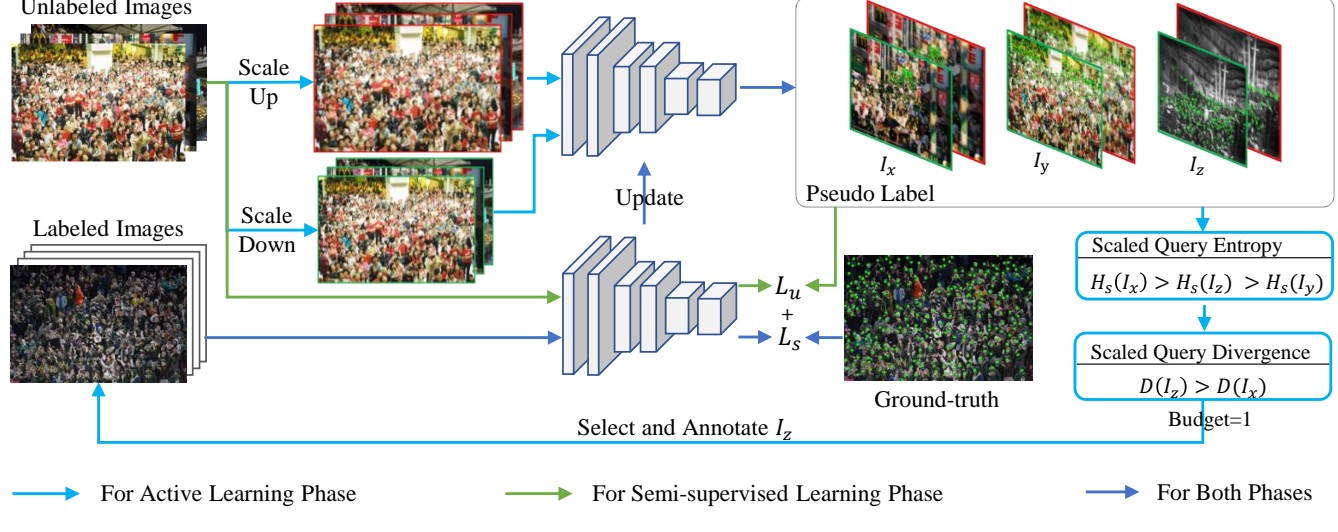

**Figure 2: The overall training pipeline of the proposed scale-based active semi-supervised learning, which consists of an active learning phase and a semi-supervised phase. The most informative unlabeled image is selected in the active learning phase for human annotation. Both labeled and unlabeled images are used to train a crowd count model in the semi-supervised phase by the supervision of ground-truth and pseudo-label. The model and the pseudo-label are progressively updated in turn.**

## 2.3 Active Learning

Active learning [2, 6–8, 31] aims to design a strategy to select data samples that can improve a previously trained model most effectively. In the field of image classification, pool-based selective sampling was adopted by many active learning methods [2, 35, 39] to select the most informative images from the unlabeled set for annotation and merge them with the labeled pool which has limited budget. For object detection, [5] leverages probabilistic modeling to estimate the uncertainty, and [51] uses multiple instance learning to select samples.

In the field of crowd counting, some researchers try to combine active learning with semi-supervised learning. AL-AC [54] was the first attempt to introduce active learning into crowd counting, which proposed a PSSW strategy to actively select samples. [25] proposed to annotate the most representative regions of an image and use GMM to make clustering. ALCrowd[34] performed approximate Bayesian inference to estimate the predictive variance to select informative images.

Existing methods simply analyze images at the image-level or patch-level. It still lacks a systematic method to learn the image at a fine-grained level.

## 3 PRELIMINARY

Denote $\mathcal{X}^{train}$ as the training set for crowd counting, all of which are not annotated at the very beginning. The training stage of active semi-supervised crowd counting can be roughly divided into two phases as the active learning phase and the semi-supervised phase.

In the active learning phase, suppose that the selected image set is $\mathcal{S}_i$ and the collection of unlabeled images is $\mathcal{U}_i$ in the $i$-th cycle satisfying $\mathcal{X}^{train} = \mathcal{S}_i \cup \mathcal{U}_i$, where $i \in 1, ..., N$ and $N$ is the total number of active learning cycles. The goal of active learning is to select a subset of the informative images under a limited annotation budget of $B$, That is to say, $B/N$ images are chosen in each cycle. For the initialization, a crowd counter $f_0$ is trained with the labeled training set $(\mathcal{S}_0, \mathcal{Y}_0)$, where $\mathcal{S}_0$ is selected randomly from $\mathcal{X}^{train}$ and $\mathcal{Y}_0$ is the crowd head point annotation by human annotators. In the $i$-th active learning cycle, the unlabeled images in $\mathcal{U}_i$ are predicted by the crowd counter $f_{i-1}$. With sample selection criteria on the predictions, a subset $\Delta\mathcal{S}_i$ with the most informative images is selected and then labeled by human annotators as $\Delta\mathcal{Y}_i$. $\mathcal{S}_i$ and $\mathcal{Y}_i$ are updated as $\mathcal{S}_i = \mathcal{S}_{i-1} \cup \Delta\mathcal{S}_i$ and $\mathcal{Y}_i = \mathcal{Y}_{i-1} \cup \Delta\mathcal{Y}_i$ respectively, which are used to train a new crowd counter $f_i$.

The semi-supervised phase is implemented after all of $N$ cycles are executed and the annotation budget $B$ is exhausted. In the semi-supervised phase, the final labeled image collection $\mathcal{S}_N$ with label $\mathcal{Y}_N$ and the unlabeled images $\mathcal{U}_N$ are used to train a final crowd count model $f$ based on various semi-supervised approaches.

Denote $\mathcal{X}^{test}$ as the test set for crowd counting, the counting result of images in $\mathcal{X}^{test}$ are predicted by the final crowd count model $f$.

## 4 METHODOLOGY

### 4.1 Overview

The scale of heads varies drastically in the crowd images, which has been demonstrated as a key factor affecting the counting performance in the fully-supervised methods [3, 26]. However, the scale factor has been neglected in active learning approaches. In this paper, we argue that an active selection strategy should also take scale as an important factor and analyze images from a fine-grained level. Hence we propose a Scale-based Active Learning (SAL) for semi-supervised crowd-counting.

The overall training pipeline of SAL is shown in Fig. 2. We use a point-based crowd counting model [40] so that the query-level prediction is achieved. In the active learning phase, we propose a simple

yet effective active labeling strategy to explicitly select informative unlabeled images guided by the intra-scale uncertainty and inter-scale inconsistency metrics. In the $i$-th active cycle, firstly we apply scaling operations to each unlabeled image $U \in \mathcal{U}_i$. Then, the counting model $f_{i-1}$ trained with all labeled images $\mathcal{S}_{i-1}$ is leveraged to infer on both upscaled image $U^{up}$ and downscaled image $U^{down}$. After that, we calculate the Scale-based Query Entropy(SQE) of images at different scales to measure the intra-scale uncertainty. Based on the rank of uncertainty, we create an initial selected pool. Finally, we propose Scale-based Query Divergence(SQD) between the upscaled and downscaled images to measure the inter-scale inconsistency. The SQD is used to determine the final selected images. SQE and SQD are detailed in Sec. 4.2 and Sec. 4.3, respectively.

Inspired by the success of pseudo-label refinement in the weakly supervised object detection [41, 43], a progressive updating scheme is proposed for the semi-supervised phase to refine the pseudo labels iteratively, which further improves the accuracy of crowd counting. The progressive updating scheme is detailed in Sec. 4.4.

## 4.2 Intra-scale Uncertainty by SQE

The point-based crowd counts model [40] generates multiple point queries for each unlabeled image. Supposing $s$ represents the down-sampling rate of the counting model and the input image is with the size of $H \times W$, the output of the model is with a size of $H_s \times W_s$, where $H_s = H/s$ and $W_s = W/s$. Each location of the output corresponds to a $s \times s$ patch of the input image. To densely detect the head points, $K$ point queries are predefined around the center of the patch. Hence the total number $T$ of point queries in the input image is $H_s \times W_s \times K$. We denote the set of point queries as $Q = \{q_j | j \in \{1, ..., T\}\}$, where $q_j$ is described with its location $(x_j, y_j)$.

In the point-based crowd counting model, a classification branch is implemented to classify each point query into the foreground (person) or background. Thus the confidence score $p_j$ of the point query $q_j$ is obtained, which indicates the probability of whether $q_j$ is a person. With this prediction probability, the entropy for the $j$-th point query can be derived as:

$$E(q_j) = -p_j log(p_j) - (1 - p_j)log(1 - p_j). \tag{1}$$

The query entropy (QE) of the image $U$ is defined as the mean entropy of all point queries, as:

$$\begin{aligned} QE(U) &= \frac{1}{T} \sum_{j=1}^{N} E(q_j) \\ &= \frac{1}{T} \sum_{j=1}^{N} (-p_j log(p_j) - (1 - p_j)log(1 - p_j)) \end{aligned}. \tag{2}$$

Higher QE indicates more information in the input image.

To ensure an image is still informative after scaling, we propose Scale-based Query Entropy (SQE) to estimate the uncertainty of images at different scales. Except for calculating the query entropy of the original image, we also input a scaled version $U_s$ of image $U$ into the counting model to calculate the corresponding query entropy $QE(U_s)$. As a result, the SQE is formulated as:

$$SQE(U) = QE(U) + QE(U_s). \tag{3}$$

In practice, the SQE can computed by different combinations of downscaled, original, and upscaled images.

With the proposed SQE, we can measure the intra-scale uncertainty of images at different scales. The images which have high SQE values are selected as informative ones. The number of selected images is slightly over the cycle budget. These selected images will be filtered again in the next step.

## 4.3 Inter-scale Inconsistency by SQD

SQE enhances the intra-scale uncertainty and creates an initial selected pool. In this section, we propose Scale-based Query Divergence(SQD) to measure the inter-scale inconsistency between the original and scaled images and decide the final selected ones.

SQD is computed on positive queries. Firstly, for a point query $q_j$ of the original image, we should find the corresponding point queries in the scaled image. In the point-based crowd counting model, a localization branch is adopted to predict the offset for each point query. Thus, the predicted localization of a point query $q_j$ of the original image is formulated as:

$$(\bar{x}_j, \bar{y}_j) = (x_j + \triangle x_j, y_j + \triangle y_j), \tag{4}$$

where $(\triangle x_j, \triangle y_j)$ is the prediction offset. We apply a scaling operation on $(\bar{x}_j, \bar{y}_j)$ with the same scale factor of the scaled image. The Euclidean distance between scaled $q_j$ and the point queries of the scaled image are computed. Denote the nearest point query for the scaled $q_j$ in the scaled image as $q_c$.

Then, KL divergence is leveraged to measure the inconsistency between the two positive point queries:

$$D(q_j, q_c) = KL(p_j || p_c), \tag{5}$$

where $p_j$ and $p_c$ are the corresponding classification score of $q_j$ and $q_c$, respectively. To measure the inter-scale inconsistency between the original and scaled images, the proposed SQD is formulated as:

$$SQD(U) = \frac{1}{O} \sum_{j=1}^{O} KL(p_j || p_c), \tag{6}$$

where $O$ denotes the number of positive queries of the original image. Since a higher SQD value represents higher inter-scale inconsistency and indicates a more scaling-sensible image, images with high SQD values are selected from the initial images pool as key images.

## 4.4 Progressive Updating Scheme

With labeled images selected by the proposed active labeling strategy, existing methods[19, 24] can be applied in the semi-supervised phase to train the final crowd counting model. However, to better refine the pseudo labels, as shown in Fig. 2, we propose a Progressive Updating Scheme (PUS) which can refine the pseudo labels iteratively for active semi-supervised crowd counting.

After $N$ cycles, we train a counting model with all of the selected labeled images $\mathcal{S}_N$ and leverage the trained model to infer on unlabeled images $\mathcal{U}_N$. The predictions are filtered by a threshold to obtain pseudo labels for $\mathcal{U}_N$. Then, we use the ground-truth of labeled images and the pseudo labels of unlabeled images as supervision to train the counting network as shown in Fig. 2. Meanwhile, the newly trained counting model is leveraged to predict the crowd counting result on the unlabeled images. As demonstrated in Fig. 2, the newly trained counting model is used to update the old one,

| Method | Type | SHA | | SHB | | QNRF | | JHU++ | | NWPU | |
|--------|------|-----|-----|-----|-----|------|-----|-------|-----|------|-----|
| | | MAE | MSE | MAE | MSE | MAE | MSE | MAE | MSE | MAE | MSE |
| MT [42] | SS | 88.2 | 151.1 | 15.9 | 25.7 | 147.2 | 249.6 | 121.5 | 388.9 | 129.8 | 515.0 |
| L2R [23] | SS | 86.5 | 148.2 | 16.8 | 25.1 | 145.1 | 256.1 | 123.6 | 376.1 | 125.0 | 501.9 |
| SUA [28] | SS | 68.5 | 121.9 | 14.1 | 20.6 | 130.3 | 226.3 | 80.7 | 290.8 | 111.7 | 443.2 |
| GP [38] | SS | 89.0 | – | – | – | 136.0 | – | – | – | – | – |
| DACount[19] | SS | 71.1 | 119.7 | 8.1 | 13.6 | 96.8 | 168.2 | 66.3 | 276.6 | – | – |
| MRL[47] | SS | 68.3 | 111.9 | 11.0 | 17.6 | 126.7 | 209.7 | 68.4 | 279.6 | 97.0 | 413.5 |
| OT-M[21] | SS | 70.7 | 114.5 | 8.1 | 13.1 | 100.6 | 167.6 | 72.1 | 272.0 | – | – |
| CU[15] | SS | 64.7 | 109.6 | – | – | – | – | – | – | – | – |
| MDC[25] | AS | 68.5 | 116.1 | – | – | – | – | 80.2 | 287.5 | 109.3 | 438.1 |
| ALCrowd[34] | AS | 66.7 | 106.8 | 9.8 | 16.1 | 95.3 | 171.5 | 64.3 | 256.7 | – | – |
| Ours | AS | **60.7** | **97.2** | **7.9** | **12.7** | **95.0** | **167.0** | **62.0** | **255.1** | **92.8** | **406.7** |

Table 1: Comparison with semi-supervised (SS) and active semi-supervised (AS) methods with 40% annotations. Bold results are the best scores and the results with an underline are the second best scores.

and the refined pseudo labels are generated by the updated model. In practice, this refinement step is executed iteratively. Through multiple refinement steps, the pseudo labels for unlabeled images are progressively refined which is beneficial for us to train a robust crowd counting model.

## 5 EXPERIMENTS

Firstly, we describe the experimental setting, which includes the datasets, implementation details, and evaluation metrics. Then, we compare the performance of our method with state-of-the-art semi-supervised and active semi-supervised crowd counting approaches. Through ablation studies, we quantitatively discuss the effectiveness of our proposed SAL strategy. Finally, we conduct experiments to discuss the impact of scale factors, every active learning cycle, and our proposed progressive updating scheme. The visualization of the prediction qualitatively verifies the efficacy of our method.

### 5.1 Experimental Setting

**Datasets.** Five open-source datasets are adopted for evaluation.

ShanghaiTech_A(SHA) [53] consists of 482 images with 244,167 annotated persons. Among these, 300 images are allocated for training purposes, while the remaining 182 images are reserved for testing. The images in the SHA dataset were randomly collected from the internet and were also utilized for ablation studies in the paper.

ShanghaiTech_B(SHB) [53] comprises 716 images with 88,498 annotated persons, indicating a lower crowd density compared to SHA. Out of these, 400 images are allocated for training, while the remaining 316 images serve as the test set.

UCF_QNRF(QNRF) [10] comprises 1535 high-resolution images with 1.25 million annotated persons. The training set consists of 1201 images, while the remaining 334 images are designated for testing. QNRF dataset presents a variety of scenes, perspectives, crowd densities, and illumination conditions, making it a highly challenging dataset for crowd counting research.

JHU-Crowd++(JHU++) [37] is a large-scale dataset consisting of 4372 images with 1.51 million annotations. 2272 images are collected for training, 500 images for validation, and 1600 images are designated for testing. Moreover, the dataset includes images with weather changes, density variations, and diverse illumination conditions which make it a challenging benchmark for crowd counting.

NWPU-Crowd(NWPU) [46] is a large-scale crowd counting dataset. It contains 3109 images for training and 500 images for validation. The number of annotated persons in a single image ranges from 0 to 20,033 and varies greatly throughout the dataset.

**Implementation details.** We optimize the network using the Adam optimizer with a learning rate of 1e-5 for the parameters of the backbone and 1e-4 for the rest of the parameters. VGG-16 [36] is utilized as the backbone. The batch size is set to 8. $K$ is set to 8 for the QNRF dataset, while set to 4 for the other datasets.

To align with the previous methods, we apply random scaling to the images, selecting a scaling factor from the range of [0.7, 1.3]. Subsequently, a random crop operation is applied to the image. The patch size is 256×256 for QNRF, JHU++, and NWPU, while 128×128 for SHA and SHB. Additionally, the patches are horizontally flipped with a probability of 0.5. To maintain the original aspect ratio, for datasets whose images are high-resolution, the longer side of each image is constrained with a hyperparameter $l$. For QNRF, $l$ is set as 1408. For JHU++ and NWPU, $l$ is set as 2048. In our proposed SAL strategy, except for the original image, we utilize one upscale factor and one downscale factor to obtain upscaled and downscaled images respectively. The scaling factors are set to 0.8 and 1.2.

For the 10% annotation budget, the first 5% labeled images are randomly initialized and 2.5% images are labeled in every active learning cycle. For the 40% annotation budget, the first 20% labeled images are randomly collected. At each cycle, after the counter is fully trained, 5% of the total images are acquired from the unlabeled set for labeling.

**Evaluation metrics.** Mean Absolute Error (MAE) and Mean Squared Error (MSE) are widely used evaluation metrics in crowd

| Method | Type | SHA | | SHB | | QNRF | | JHU++ | | NWPU | |
|---|---|---|---|---|---|---|---|---|---|---|---|
| | | MAE | MSE | MAE | MSE | MAE | MSE | MAE | MSE | MAE | MSE |
| MT [42] | SS | 94.5 | 115.5 | 15.6 | 24.5 | 145.5 | 250.3 | 90.2 | 319.3 | 144.1 | 508.6 |
| L2R [23] | SS | 90.3 | 115.5 | 15.6 | 24.4 | 148.9 | 249.8 | 87.5 | 315.3 | 138.3 | 550.2 |
| IRAST [24] | SS | 86.9 | 148.9 | 14.7 | 22.9 | 135.6 | 233.4 | 86.7 | 303.4 | – | – |
| DACount[19] | SS | 82.5 | 123.2 | 10.9 | 19.1 | 115.1 | 193.5 | 74.0 | 297.1 | – | – |
| MRL[47] | SS | 80.2 | 125.6 | 12.1 | 19.7 | 132.5 | 221.2 | 80.1 | 299.9 | 132.9 | 511.3 |
| OT-M[21] | SS | 80.1 | 118.5 | 10.8 | 18.2 | 113.1 | 186.7 | 73.0 | 280.6 | – | – |
| CU[15] | SS | 70.8 | 116.6 | **9.7** | 17.7 | **104.0** | **164.3** | 74.9 | 281.7 | 108.8 | 458.0 |
| AL-AC[54] | AS | 87.9 | 139.5 | 12.7 | 20.4 | 131.4 | 229.7 | – | – | – | – |
| MDC[25] | AS | 79.6 | 127.5 | 12.7 | 20.3 | 128.6 | 226.4 | – | – | – | – |
| Ours | AS | **69.7** | **114.5** | **9.7** | **17.5** | 106.7 | 171.3 | **69.7** | **263.5** | **107.1** | **443.6** |

Table 2: Comparison with semi-supervised (SS) and active semi-supervised (AS) methods with 10% annotations. Bold results are the best scores and the results with an underline are the second best scores.

counting which is also adopted in this paper. With the predicted count $C_i$ of the $i$-th image and the ground truth count $GT_i$. MAE and MSE can be formulated as follows:

$$MAE = \frac{\sum_{i=1}^{N} |C_i - GT_i|}{N} \quad (7)$$

$$MSE = \sqrt{\frac{\sum_{i=1}^{N} |C_i - GT_i|^2}{N}} \quad (8)$$

where $N$ is the total number of testing images.

## 5.2 Comparison with State-of-the-art Methods

In this section, we compare our proposed method with state-of-the-art semi-supervised and active semi-supervised methods.

The results with 40% annotations are shown in Table 1. It shows that our approach outperforms other state-of-the-art methods on all datasets, regardless of the type of methods. Compared to the second-best results, our method reduces the MAE by 4.0, 0.2, 0.3, 2.3, and 4.2 points on datasets SHA, SHB, QNRF, JHU++, and NWPU, respectively. Compared to the best active semi-supervised learning method ALCrowd [34], our method reduces the MAE by 6.0, 1.9, 0.3, and 2.3 points on datasets SHA, SHAB, QNRF, and JHU++, respectively. On the NWPU dataset, we achieve about 15% improvement of MAE compared with the state-of-the-art active semi-supervised learning method.

Our method is also effective when the labeling budget is lower. As shown in Table 2, with only 10% annotations available, our approach improves all the previous methods by at least 1.1, 3.3, and 1.7 MAE on datasets SHA, JHU++, and NWPU respectively. Moreover, compared to the best active learning method MDC [25], the improvement of MAE is 9.9 points on SHA, 3.0 points on SHB, and 21.9 points on QNRF. These outstanding results demonstrate the effectiveness of our method in the field of active semi-supervised crowd counting.

| Method | MAE | MSE |
|---|---|---|
| Random | 68.1 | 115.5 |
| ALCrowd[34] | 66.7 | 106.8 |
| QE | 66.2 | 109.8 |
| SQE | 62.3 | 104.5 |
| SQD | 63.6 | 105.2 |
| SQE + SQD | **60.7** | **97.2** |

Table 3: Ablation study of different active selection strategies on SHA dataset.

## 5.3 Ablation Study

We conduct ablation experiments to verify the effectiveness of our proposed intra-scale uncertainty (SQE) and inter-scale inconsistency (SQD). The results are shown in Table 3.

All experiments are conducted on the SHA dataset with a 40% annotation budget. We start with random sampling on a progressive updating scheme. It achieves 68.1 MAE and 115.5 MSE. While the state-of-the-art active semi-supervised method ALCrowd [34] achieves 66.7 MAE and 106.8 MSE which is better than random sampling, indicating that active labeling is important for semi-supervised crowd counting. With the application of QE which does not include a scaling operation, it improves the performance compared to random sampling. But it only slightly improves the MAE and obtains a worse MSE compared to ALCrowd [34]. We believe that this limited improvement is because the intra-scale uncertainty cannot be accurately estimated by simply calculating QE and overlooking the impact of scale.

The proposed SQE is adopted to study its efficiency. With SQE which includes scaling operations, its performance significantly exceeds using QE, the error reduces by 3.9 and 5.3 for MAE and MSE, respectively. Hence, the results verify that the proposed SQE can accurately estimate the intra-scale uncertainty and select informative images. By measuring the inter-scale inconsistency between

images at different scales, images that are sensitive to scale variation are selected. As the results shown in Table 3, SQD improves the performance by 4.5 MAE compared with the random selection. It proves the effectiveness of inter-scale inconsistency. Finally, the combination of SQE and SQD enhances the overall performance even further, improving 6.0 for MAE and 9.6 for MSE compared to ALCrowd [34], suggesting that the proposed SQE and SQD are complementary.

## 5.4 Discussions

**The impact of scale factors.** In the proposed SAL, with the original image, two scale factors are utilized to produce their corresponding downscaled and upscaled image respectively. We conduct experiments to study the effect of different combinations of scale factors. All experiments are performed on the SHA dataset with a 40% budget and the comparison results are summarized in Table 4.

The results show that all three scale factor combinations can improve the performance compared to random sampling. The best performance is obtained when the scale factors are set to 0.8&1.2, achieving 60.7 MAE and 97.2 MSE. In specific, when the scale factors are set to 0.9&1.1, the small differences between two images can lead to a decrease in diversity, making performance unsatisfactory. When the scale factors are 0.7&1.3, the differences are too big to estimate accurate intra-scale uncertainty and inter-scale inconsistency, which also causes a decline in performance.

**The impact of active labeling cycle.** We conduct experiments on the SHA dataset to examine the impact of every active learning cycle. The labeling budget is set to 40%. The first 20% images are randomly sampled, and the rest are sampled in four cycles by our proposed active labeling strategy. The performance of every cycle is compared to random sampling, and the comparison results are shown in Table 5.

Compared to random sampling, our method which leverages SAL strategy to actively select images has consistent promotion in all cycles. As shown in Table 5, the improvement will be larger when the labeled percentage becomes bigger. This is reasonable since as the number of cycles increases, more images selected by our proposed SAL strategy are annotated. This quantitative experiment proves that our SAL strategy can select informative images to improve performance.

**The effect of the progressive updating scheme.** In the semi-supervised phase, a progressive updating scheme(PUS) is proposed to refine the pseudo labels iteratively. In Table. 6 and Table. 7, we conduct experiments to analyze the effect of different iterations and verify the superiority of PUS against the existing semi-supervised method. All experiments are conducted on SHA with an annotation budget of 40%.

As shown in Table. 6, when the number of iterations increases, the MAE first decreases quickly in the first two iterations and then begins to oscillate. For efficiency, 2 iterations are applied in our other experiments. Compared to the baseline (0 iteration), the performance is largely improved by 5.2 MAE, which proves that our iterative refinement for pseudo-labels is effective. Meanwhile, the significant improvement in the first two iterations indicates that the converging speed of our PUS is fast and the efficiency is excellent.

| Scale factors | MAE | MSE |
|---|---|---|
| 0.9 & 1.1 | 63.5 | 103.6 |
| 0.8 & 1.2 | **60.7** | **97.2** |
| 0.7 & 1.3 | 62.0 | 102.7 |

**Table 4: The impact of different scale factors.**

| Percentage | Ours | | Random | |
|---|---|---|---|---|
| | MAE | MSE | MAE | MSE |
| 20% | 72.5 | 132.3 | 72.5 | 132.3 |
| 25% (1st cycle) | 68.4 | 120.3 | 71.1 | 127.4 |
| 30% (2nd cycle) | 65.2 | 112.6 | 69.8 | 122.9 |
| 35% (3rd cycle) | 62.6 | 104.7 | 68.8 | 118.6 |
| 40% (4th cycle) | 60.7 | 97.2 | 68.1 | 115.5 |

**Table 5: The impact of active labeling cycle.**

| Iterations | 0 | 1 | 2 | 3 | 4 |
|---|---|---|---|---|---|
| MAE | 65.9 | 62.2 | 60.7 | 61.1 | 60.8 |
| MSE | 113.8 | 105.1 | 97.2 | 98.6 | 96.1 |

**Table 6: The results of PUS with different iterations.**

| Method | MAE | MSE |
|---|---|---|
| DACount[19]+Random | 71.1 | 119.7 |
| DACount[19]+SAL | 67.3 | 114.3 |
| PUS+Random | 68.1 | 115.5 |
| PUS+SAL | **60.7** | **97.2** |

**Table 7: The effect of the progressive updating scheme.**

In Table. 7, we choose the previous state-of-the-art semi-supervised method DACount [19] for comparison. We train DACount [19] and our proposed PUS with samples selected by random sampling (+Random) and SAL strategy (+SAL). As the results demonstrate, the performance of PUS surpasses DACount whether using random sampling or SAL strategy. Our PUS which can consistently refine the pseudo labels shows its effectiveness. Also, it is worth noting that the performance of DACount+SAL outperforms DACount+Random which strongly validate the superiority of our proposed active labeling strategy and the representativeness of our selected images.

## 5.5 Visualization

To examine the impact of our proposed SAL strategy qualitatively, we visualize some results predicted by our model and DACount[19] in Fig. 3. All models are trained with 40% labeled images. As shown in the third column of Fig. 3, the single use of SQE has already resulted in improvement compared to random sampling. Moreover, the combination of SQE and SQD which can accurately measure the

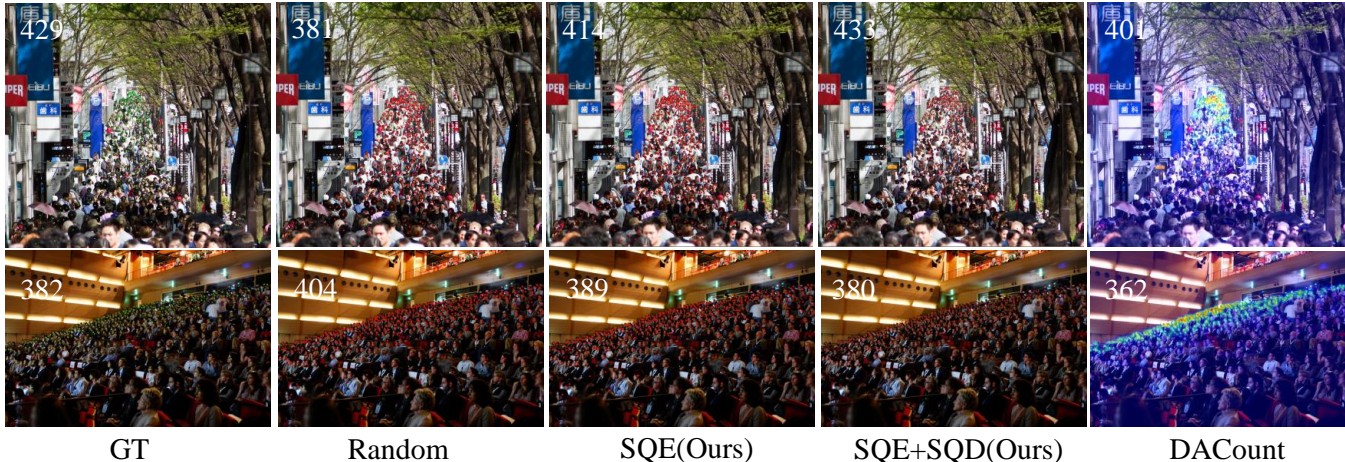

**Figure 3: Visualization of the crowd counting predictions. All models are trained with 40% annotations. Predicted counting results are shown at the top-left corner. Random, SQE, and SQE+SQD indicate the labeled samples are selected randomly, with SQE, and with both SQE and SQD, respectively. DACount [19] is a SOTA semi-supervised method.**

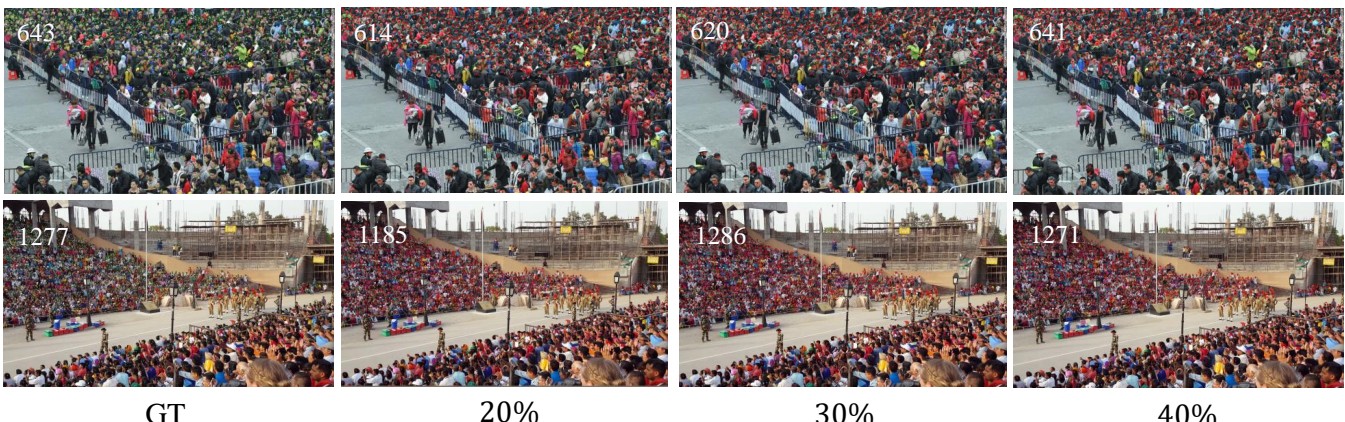

**Figure 4: Visualization of the results of model trained with 20%, 30% and 40% labeled images in active learning phase. The accuracy is improved following the increasing percentage of annotation.**

intra-scale uncertainty and inter-scale inconsistency forms complementarity and further improves the performance as the fourth column of Fig. 3 shows. Meanwhile, our proposed method achieves a better performance compared to DACount[19].

We also visualize the predictions of our approach trained with 20%, 30%, and 40% labeled images in Fig. 4. We first annotate 20% images that are randomly selected. The results of a model trained with 20% images are shown in the second column. We can find that the errors are a little bit large. Then we leverage the SAL strategy to actively select another 10% and 20% images to annotate and show the results in the third column and fourth column. The comparison between the second, third, and fourth columns shows that our proposed SAL strategy can largely improve the performance of a counting model. Furthermore, the more images are actively selected, the bigger improvement is obtained.

## 6 CONCLUSION

We present a simple yet effective scale-based active learning strategy for active semi-supervised crowd counting in this paper. For active semi-supervised crowd counting, it is the first attempt to analyze images from the query-level and take scale as an important factor in informative sample selection. Specifically, in the active learning phase, intra-scale uncertainty and inter-scale inconsistency are proposed to actively choose images step by step. In addition, in the semi-supervised phase, to iteratively refine the pseudo labels for unlabeled images, a progressive updating scheme is introduced. Extensive experiments conducted on public crowd counting datasets provide strong evidence of the effectiveness of the proposed method. In the future, it's worth analyzing the combination and incorporation of the proposed scale-based active labeling strategy and other active semi-supervised crowd counting approaches.

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
