# OpenReview forum: "Boosting Semi-supervised Crowd Counting with Scale-based Active Learning"
_acmmm.org/ACMMM/2024/Conference — MM2024 Poster_

### Official Review · Reviewer_d21j · 2024-05-16

**Rating:** 4
**Confidence:** 4

**Summary:**

In this work, the authors study how to use active learning techniques for semi-supervised crowded counting. The core of this work is to select informative images to label. The inter-scale uncertainty and inter-scale inconsistency information are used to achieve this goal. Furthermore, a part set of unlabeled images can be annotated. Then, the labeled images and the unlabeled images are employed to train a crowd-counting model. The authors conduct extensive experiments on four public datasets, including ShangHaiTech, QNRF, JHU++, and NWPU. The experimental results show the effectiveness of this method.

**Strengths:**

+ How to use active learining in the filed crowd counting is a practical issue.
+ The framework of the proposed method is elegent.
+ The experimental results of this work are impressive.

**Limitations:**

- The selection process is not clear. Only metrics are introduced (entropy in SQE, KL divergence in SQD).
- what is the meaning of "positive queries" in line 418?
- It would be better if the visualization results of selected images were provided. In addition, the predicted crowd-counting results can be presented as density maps.
- What is the annotation budget? Is this the number of annotated images or the number of point clicks?
- This work conducts the experiments only on a VGG-based backbone, it would be better if another backbone, for example, a transformer-based backbone is employed to verify the effectiveness of this method.

**Suitability:**

3

---

### Official Review · Reviewer_TUaP · 2024-05-18

**Rating:** 3
**Confidence:** 3

**Summary:**

This paper studies the task of crowd counting. It proposes two novel selection criteria for active semi-supervised crowd counting, which are scale-based query entropy to measure the intra-scale uncertainty, and scale-based query divergence to measure the inter-scale inconsistency. It also implements a progressive updating scheme, in which the pseudo-labels for unlabeled images are refined iteratively.

**Strengths:**

1. The proposed two criteria are simple and effective to tackle the neglect of  scale factor in active learning.
2. Experiemt results on benchmark datasets validates the effecttiveness of the proposed method.
3. The writting and organization is fluent.

**Limitations:**

1. The key contribution of this work is two selection criteria, which makes the novelty of this paper somewhat limted.
2. The process illustrated by Fig.2 is confusing. It needs to show the orders of different process. And what is Ix, Iy, Iz? What do  the "Hs(Ix)>Hs(Iz)>Hs(Iy)" and the "D(Iz)>D(Ix)" mean?
3. Current visualization results are meaningless. The authors need to figure out another way to visually evaluate the prediction results.
4. It is better to add more ablation studies on different datasets.
5. Gramma mistakes need to be fixed, such as in line 404: can computed -> can be computed.

**Suitability:**

3

---

### Official Review · Reviewer_4npx · 2024-05-23

**Rating:** 3
**Confidence:** 3

**Summary:**

This paper proposes a simple yet effective active labeling strategy for active semi-supervised crowd counting. Specifically, an intra-scale uncertainty metric and an inter-scale inconsistency metric are proposed to facilitate the selection of the most informative images for annotation. The intra-scale uncertainty is quantified through the sum of the query-level entropy of images at different scales. The Inter-scale inconsistency is measured by the divergence between the query-level predictions of upscaled and downscaled images. Besides, a progressive updating scheme is proposed to refine the pseudo-labels iteratively during training. Extensive experiments demonstrate the effectiveness of the method.

**Strengths:**

1. The motivation behind this paper is sound, and the technical correctness of the work is well demonstrated.
2. This paper is well organized and the experimental results are relatively significant.

**Limitations:**

1. Fig.1 has very little information. And it is hard to know what is “intra-scale” and “inter-scale” from this description.
2. The word of “Budget=1” in Fig.2 is hard to understand.
3. It seems that the proposed Progressive Updating Scheme (PUS) is just to update the models online, which lacks technicality to support as a contribution.
4. This paper lacks a comparison of the latest work. For example, “Calibrating uncertainty for semi-supervised crowd counting (ICCV 2023) [1]”, which achieves MAE=70.76, RMSE=116.92 on SHA, MAE=9.71 and RMSE=17.74 on SHA, and MAE=104.04 and RMSE=164.25 on QNRF. These results are either comparable to or superior to the performance of this study. Notably, It even does not introduce active learning to select the data to be labeled.
5. There is a discrepancy between the content and caption in Tab. 7. The table presents results for the SAL active selection strategy; however, the caption incorrectly describes the effect of the progressive updating scheme.

[1] Li, Chen, et al. "Calibrating uncertainty for semi-supervised crowd counting." 2023 IEEE/CVF International Conference on Computer Vision (ICCV). IEEE, 2023.

**Suitability:**

3

---

### Official Review · Reviewer_2AME · 2024-05-26

**Rating:** 4
**Confidence:** 4

**Summary:**

This paper proposes a novel approach to improve semi-supervised crowd counting by integrating scale-based active learning strategies. The method focuses on intra-scale uncertainty and inter-scale inconsistency to select the most informative samples for annotation, addressing the variability in head sizes within crowd images. Additionally, a progressive updating scheme is implemented to iteratively refine pseudo-labels, further enhancing the model's performance.

**Strengths:**

1) The introduction of scale-based metrics (intra-scale uncertainty and inter-scale inconsistency) for sample selection in active learning is innovative and addresses a significant gap in the current methods.

2) The approach is highly relevant for practical applications where annotation costs are a concern, making semi-supervised methods particularly valuable.

3) The results are robust, showing significant improvements in Mean Absolute Error (MAE) and Mean Squared Error (MSE) across different datasets and annotation budgets.

**Limitations:**

1) For the related work, the authors should discuss the weakly and unsupervised methods[1,2]
2) Tab.1 should compare with more recent SOTA methods[3,4,5].
3) Although the paper includes ablation studies to demonstrate the effectiveness of intra-scale uncertainty and inter-scale inconsistency, further experiments isolating each component's impact on different datasets would provide deeper insights.

[1] crowdclip unsupervised crowd counting via vision-language model. CVPR 23.
[2] Completely Self-Supervised Crowd Counting via Distribution Matching. ECCV 22.
[3] Focal Inverse Distance Transform Maps for Crowd Localization. TMM 22.
[4] Point-Query Quadtree for Crowd Counting, Localization, and More. ICCV 23.
[5] Diffuse-Denoise-Count: Accurate Crowd-Counting with Diffusion Models.

**Suitability:**

2

---

### Meta-Review · Area_Chair_JrDP · 2024-06-30

**Recommendation:** Accept (Poster)
**Confidence:** 5

**Metareview:**

This paper received one borderline accept, one borderline reject and one borderline accept final ratings from the reviewers. The borderline reject reviewer main concern is that the experimental results in this paper are not sufficiently significant. But the current results can sufficiently confirm the effectiveness of the proposed approach. AC agrees that this paper benefits from good writing and interesting idea. However, the authors are encouraged to make the necessary changes to the best of their ability.